# Relationship Between Chronic Wasting Disease (CWD) Infection and Pregnancy Probability in Wild Female White-Tailed Deer (*Odocoileus virginianus*) in Northern Illinois, USA

**DOI:** 10.3390/pathogens14080786

**Published:** 2025-08-07

**Authors:** Jameson Mori, Nelda A. Rivera, William Brown, Daniel Skinner, Peter Schlichting, Jan Novakofski, Nohra Mateus-Pinilla

**Affiliations:** 1Illinois Natural History Survey-Prairie Research Institute, University of Illinois Urbana-Champaign, Champaign, IL 61820, USA; jmori@illinois.edu (J.M.); river@illinois.edu (N.A.R.);; 2Illinois Department of Natural Resources, Division of Wildlife Resources, Springfield, IL 62702, USA; 3Department of Animal Sciences, University of Illinois Urbana-Champaign, Champaign, IL 61801, USA; 4Department of Pathobiology, University of Illinois Urbana-Champaign, Urbana, IL 61802, USA; 5Department of Natural Resources & Environmental Sciences, University of Illinois Urbana-Champaign, Urbana, IL 61801, USA

**Keywords:** cervid, *Cervidae*, prion, transmissible spongiform encephalopathy, TSE, chronic wasting disease, CWD, *Odocoileus virginianus*, reproduction

## Abstract

White-tailed deer (*Odocoileus virginianus*) are a cervid species native to the Americas with ecological, social, and economic significance. Managers must consider several factors when working to maintain the health and sustainability of these wild herds, including reproduction, particularly pregnancy and recruitment rates. White-tailed deer have a variable reproductive capacity, with age, health, and habitat influencing this variability. However, it is unknown whether chronic wasting disease (CWD) impacts reproduction and, more specifically, if CWD infection alters a female deer’s probability of pregnancy. Our study addressed this question using data from 9783 female deer culled in northern Illinois between 2003 and 2023 as part of the Illinois Department of Natural Resources’ ongoing CWD management program. Multilevel Bayesian logistic regression was employed to quantify the relationship between pregnancy probability and covariates like maternal age, deer population density, and date of culling. Maternal infection with CWD was found to have no significant effect on pregnancy probability, raising concerns that the equal ability of infected and non-infected females to reproduce could make breeding, which inherently involves close physical contact, an important source of disease transmission between males and females and females and their fawns. The results also identified that female fawns (<1 year old) are sensitive to county-level deer land cover utility (LCU) and deer population density, and that there was no significant difference in how yearlings (1–2 years old) and adult (2+ years old) responded to these variables.

## 1. Introduction

White-tailed deer (*Odocoileus virginianus*) are a species of wild ungulate in the family *Cervidae* native to the Americas [1]. These deer are a keystone species that influence the success of plants in their habitat through preferential browsing [2], impact the fauna living alongside them [3], and have long been a source of food, materials, and sport for both indigenous and non-indigenous peoples [4]. The impact of this species is felt on multiple fronts, including its contribution to vector-borne diseases [5], crop depredation [6], and deer–vehicle collisions [7]. The magnitude of this impact is directly proportional to population size, which itself depends on the herd’s reproductive capacity. In Illinois, the white-tailed deer mating season is from October to January, and offspring are born between April and July, with most fawns born in May or June [8]. Analysis of data from female deer culled in northern Illinois showed that 66% were pregnant, including 20.5% of fawns (<1 year old), 77.6% of yearlings, and 85.8% of adults (2+ years old) [9]. Female white-tailed deer can produce a litter every year, with litter sizes ranging from one to five offspring, but averaging 1.2 fetuses for fawns and 2.0 for adults [9]. The life span of an individual white-tailed deer differs based on habitat, hunting, and sex, with male deer in the U.S. state of Illinois living an average of 2.5 years and females an average of 5.5 years [10], which means that an average female deer produces 8–9 offspring in her lifetime.

There are many factors that influence reproductive success in white-tailed deer. Age is a primary factor, as both males and females can breed as fawns; however, their odds of success increase with age [1]. Maturity in reproductive organs improves the odds of conception [11], and older deer tend to have more and larger offspring per litter [9] and are more successful at raising offspring to independence [12]. Body weight also impacts reproduction, so that females with a higher mass are more likely to reproduce and have larger litters [13].

Deer habitat plays a role in reproductive success, where areas with better deer habitat support healthier and more reproductively capable white-tailed deer [14]. In the U.S. Midwest, deer habitat includes a matrix of forests, grasslands, agriculture, and wetlands [15], with the presence of edges between different land cover types that are preferred by white-tailed deer and the abundance of nutritious crops at least partly explain the region’s high fawn pregnancy rates of 20.5% in Illinois [9] and up to 30% in Iowa and Nebraska [16]. Deer population density also influences whether female deer—especially younger deer—get pregnant [17], with deer density commonly approximated using hunter-harvest data [18].

While the reproductive capacity of white-tailed deer is notably high in the agricultural Midwest, some factors threaten white-tailed deer populations, such as chronic wasting disease (CWD) [19]. Present in wild and captive cervid populations in the U.S., Canada, Norway, Finland, Sweden, and South Korea, chronic wasting disease is a transmissible spongiform encephalopathy (TSE) of cervids with a 100% fatality rate [19]. CWD has a long incubation period, typically lasting 2–4 years, during which deer shed CWD infectious proteins in body fluids and excreta [20]. Horizontal transmission is the most common mechanism of disease spread, but the infectious material shed by the deer can contaminate the environment and facilitate indirect transmission [19], thus the need to decrease the number of infectious animals in the landscape. Furthermore, there is evidence that vertical transmission occurs in white-tailed deer [21], as it does in other cervid species [22]. CWD prions have been detected in reproductive tissues [23], saliva [24], mucous [25], and blood [24], so activities involving those tissues are a high risk for transmission, such as breeding and the care of offspring, which involves a lot of nursing and grooming [26]. Due to the long incubation period, even if a female deer is born infected (in utero transmission) or contracts CWD soon after birth, that animal could still produce multiple litters while infected, thereby contributing to local transmission. Thus, CWD could have serious population-level effects if the reproductive process is a primary mechanism of CWD transmission and spread.

Observational [27,28,29] and modeling studies [30,31] project population declines should CWD spread uncontrolled. Lower survival rates mean female deer have less time to produce offspring, reducing their contributions to the population. In the interest of preserving this important species, it is therefore essential to build our understanding of the “how”, “why”, “when”, “where”, and “what” of deer reproduction and the potential impact of CWD.

Despite the need, there are few studies on this subject. One study examining the impact of maternal CWD infection on offspring in utero found that fetuses of CWD-positive deer weighed 1% less in the second trimester of pregnancy than those from CWD-negative deer, with potentially greater impacts at birth [32]. Another study found that CWD-positive female deer were more likely to be parents than CWD-negative deer—possibly due to the sampling bias associated with increased vulnerability to harvest—and that affected deer may be less able to care for their offspring [33]. However, not all studies agree that CWD influences reproduction to a meaningful degree. A modeling study in mule deer (*Odocoileus hemionus*) concluded that CWD infection did not impact pregnancy probability or fawn recruitment [30]. One longitudinal study of ranched elk (*Cervus canadensis*) saw no impact of CWD on pregnancy in those animals [34], while another found that CWD slowed elk population growth [35].

Studies of the impact of malnutrition—the proposed mechanism of altered pregnancy probabilities in CWD-infected animals—in other mammals suggest that pregnancy rates are not often impacted by malnutrition [36,37,38]. However, there are differences in how pregnancy can be affected, depending on the type of insult (e.g., environmental, nutritional, illness), the duration, and timing. Stress during pregnancy has been shown to cause low birth weight, inefficient growth, death (neonatal demise and stillbirth), and metabolic deficiencies in ruminants [37]. Thorson and Prezotto (2024) [39] demonstrated that protracted maternal malnutrition in cows leads to an increase in urea nitrogen concentration and observed a difference in the concentration of metabolites from the dam to neonatal cerebrospinal fluid, which may affect central nervous system development.

Research conducted on other prion diseases, such as scrapie (sheep) and bovine spongiform encephalopathy (BSE; cattle), found that the offspring of scrapie-infected sheep [38,40] and BSE-infected cows [41] were at a higher risk of developing these prion diseases themselves, even when the impact of genetics was controlled for, and that scrapie-affected sheep produced fewer offspring [42]. These observations reinforce concerns about similar outcomes in cervids.

The limited nature of our current understanding of this subject in cervids prompted our investigation. Illinois’ first documented cases of CWD were reported in November of 2002 in three north-central counties—Boone, Winnebago, and McHenry—and the disease has since expanded to the southeast and southwest [43]. An increase in CWD prevalence has also been observed over time [43]. As CWD affects more locations at higher intensities, the associated risks also increase, raising the question of whether the population declines observed or predicted in other states will also begin to affect Illinois [27,29,30,31]. It was therefore our aim to determine whether pregnancy probability in the Illinois wild white-tailed deer herd has been affected by CWD to date by performing multilevel Bayesian logistic regressions to relate pregnancy probability to individual, environmental, and anthropogenic factors. We found that CWD does not impact pregnancy probability, but that the relationship between deer and their environment varies depending on the age of the animal.

## 2. Materials and Methods

### 2.1. Software

All analyses were conducted using R software (v. 4.4.3) [44]. All datasets generated in this study are publicly available at the Illinois Data Bank (https://doi.org/10.13012/B2IDB-6395643_V1).

### 2.2. Reproductive Dataset and Additional Covariates

The Illinois Department of Natural Resources (IDNR) has tested wild white-tailed deer (*Odocoileus virginianus*) for chronic wasting disease (CWD) since it was first detected in northern Illinois in fiscal year 2003, with a fiscal year being the time between July 1st of one calendar year and June 30th of the next. We use data collected during CWD surveillance and the IDNR’s locally focused culling events, which aim to reduce the local deer population density and thereby slow CWD spread [45]. The data collected includes CWD status, age, sex, location, kill date, and pregnancy status of female deer in CWD-infected areas. Starting in fiscal year 2017, IDNR also began measuring maternal body weight at some locations. These data can thus be viewed as two separate, but related, datasets, with one omitting (FY2003–2024) and one including (FY2017–2024) maternal body weight.

The dataset containing maternal weight data had observations for 2071 female deer that were pregnant and 2957 that were not. There were 1304 fawns (<1 year), 719 yearlings (1–2 years old), and 3005 adults (2+ years). The dataset that excluded maternal body weight contained 5673 pregnant and 4110 non-pregnant deer; 2524 were fawns, 1484 were yearlings, and 5775 were adults. A total of 541 TRSs (township, range, and section), 141 townships, and 19 counties were included in the dataset lacking maternal weight. It should be noted that culling occurs between January and March—after hunter harvest—and so fawns are 8–10 months old, which helps to explain the detection of CWD in 27 fawns in the FY2003–2024 dataset. Fawns are not tested when harvested during hunting season, because at this time the fawns are about 6 months old and, due to the long incubation period and slow accumulation of the prion protein that allows diagnosis [19], CWD is rarely identified in this age group.

Using the available spatiotemporal information, values for other covariates of interest were linked to each female deer. A summary of the deer, deer population, and habitat covariates chosen for modeling pregnancy probability is provided in Table 1. Spatially explicit covariates were assessed at the TRS (mean = 2.6 km^2^), township (mean = 83 km^2^), and/or county levels (mean = 1431 km^2^).

Four characteristics of the individual female deer were included in this analysis. Maternal age was determined by trained IDNR biologists based on tooth replacement and wear [46], with animals classified as fawns (<1 year old), yearlings (1–2 years), or adults (2+). A categorical age was used instead of numerical, because the dataset more consistently used this means of classifying age, and therefore expanded the usable sample size. Age has been well-established as a critical factor in reproduction [9], as has maternal body weight [13], also included as a covariate in this study. Maternal infection with chronic wasting disease (CWD) was the third individual-level parameter used in the models, with one hypothesis being that the disease may alter the probability of pregnancy through malnutrition or loss of body condition.

The last of the individual-level covariates was the day of the fiscal year on which the female deer was culled. The fiscal year was used instead of the calendar year because it allows reproduction (Fall), deer culling for CWD management (Winter), and fawning (late Spring) to be included in the same time period. The day of the fiscal year is thus the fiscal year equivalent of calendar day of the year, with day 1 corresponding to July 1st and day 365 to June 30th. Fiscal year days were included in this analysis because deer culled later in the fiscal year have larger fetuses due to a longer gestational period than deer culled earlier in the fiscal year.

The deer land cover utility score (LCU) was a metric developed by Mori et al. (2024) to score the quality and quantity of deer habitat in an area using data from the National Land Cover Database. Land classes considered deer habitat were forests, wetlands, grasslands, shrubs, pastures, and row crops [15]. Higher LCU scores indicated areas with more abundant and better-quality deer habitat [15]. These scores were utilized in these models to test whether the deer LCU at the TRS, township, or county levels best explained the relationship between pregnancy and habitat. Lastly, deer population density was included in the models to account for population-associated pressures, particularly in younger deer [17], and was estimated by the density of hunter-harvested deer in a county [18]. Hunter-harvested data include both male and female deer of all age groups, with a roughly 50/50 split by sex [47].

All continuous covariates were mean-centered and divided by twice their standard deviation, while the binary covariate (CWD) was mean-centered to put all covariates on the same scale for interpretation [48].

### 2.3. Multilevel Bayesian Logistic Regression

Multilevel Bayesian logistic regressions were constructed using the “brms” package in R [49]. All models used a “Bernoulli” distribution with a “logit” link function and 10,000 net iterations. The default priors of the “brm” function in the “brms” package (v. 2.22.0) were used; improper uniform priors for fixed effects and a half student-t prior for random effects [49]. Modeling was conducted first for the dataset that included maternal weight to determine if this variable was significant and necessary to understand pregnancy probability in white-tailed deer. If maternal weight was not statistically significant, the models would be rerun with the dataset that omitted maternal weight. Statistical significance was defined as the regression coefficient having a 95% credibility interval that did not contain 0. This credibility interval is the Bayesian alternative to the 95% confidence interval. Multiple, nested spatial levels—TRS, township, and county—were included in the model structure to account for deer being more similar to other deer in the same location and these levels were further nested in the temporal level “fiscal year” (FY) to obtain the level structure “FY/County/Township/TRS” to account for similarities between deer culled in the same time period.

The first modeling step was to construct “null” models that contained only the outcome (pregnancy status) and the levels (FY/County/Township/TRS) to determine how much of the variance in the data was explained by the model structure alone. Next, models were run with all the covariates (Table 1) and potential interaction terms between those covariates. Insignificant interaction terms were then removed from the model, and the model was rerun. Multicollinearity was tested for by using the “performance” package (v0.15.0) in R [50], since it can cause issues with regression results. Vehtari defines low multicollinearity as having a variance inflation factor (VIF) < 5, moderate as being 5 < VIF < 10, and high as VIF > 10. Covariates that demonstrated moderate or high multicollinearity were tested in separate models to determine which best explained the data based on the lowest leave-one-out cross-validation information criterion (LOOIC) score [51]. The models with the best fits were then used to understand the relationships between pregnancy probability and the tested covariates. To aid in interpretation, each regression coefficient was converted to be the percentage that the pregnancy probability changes when the covariate goes from a low to a high value [48] using Equation (1). A negative percentage means that an increase in the covariate decreases the probability of pregnancy, while a positive percentage increases it.
[EXP (coefficient) − 1] × 100](1)

All models were checked for convergence using the Rhat statistic, which has a minimum and maximum very close to 1 when the model has converged [44]. The sample size was assessed by calculating the minimum effective sample size for the model and comparing it to the threshold of 0.1, with values larger than this threshold indicating an adequate number of samples [52]. The Tjur’s R^2^ was calculated to quantify how much the models explained the amount of variance in the data, with this version of the standard R^2^ statistic used because the model was a logistic regression [53].

## 3. Results

The models run with data collected between fiscal years 2017 and 2024 determined that maternal age and weight were moderately collinear (VIF = 9.22), which is above the suggested threshold of 5 [54]. Modeling maternal age and weight together and comparing the outputs yielded the regression coefficients for these covariates and their 95% credibility intervals, with each model’s LOOIC score reported in parentheses (Table 2). The maternal age category “fawn” was always significant, while the age category of “yearling” and maternal weight were never significant. These findings facilitated the conclusion that maternal weight was not essential for understanding pregnancy probability in northern Illinois wild white-tailed deer (Table 2), and so all subsequent analyses were performed with the dataset excluding maternal weight.

Once this conclusion was reached, modeling with reproductive data collected between fiscal years 2003 and 2024—which did not include maternal weight—was carried out. All models converged with an Rhat around 1 and a sufficiently large effective sample size ratio of ≥0.1 [52].

Figure 1 visualizes the data used in this analysis. Figure 1A shows the locations of the counties from which female deer were culled and examined for pregnancy, with the number of deer examined for pregnancy listed beside the county name in parentheses. Figure 1B shows the cumulative number of CWD cases from all harvest and management sources by TRS in the counties studied between fiscal years 2003 and 2024, with darker colors and larger symbols indicating areas with a higher case burden. Figure 1C shows the number of female deer examined for pregnancy during the IDNR’s locally focused culling efforts in the fiscal years studied, by TRS, with larger symbols and darker colors representing a higher number of deer. Comparison between Figure 1B,C shows that the sample sizes of female deer were roughly proportional to the number of detected CWD cases, which is because culling efforts are targeted at or near TRSs with more CWD.

The Tjur’s R^2^ for the FY2003–2024 null model was 0.36, indicating that 36% of the variance in the data was explained just by the levels. Individual deer, environmental, and deer population covariates and their interactions were then added to this null model to explore their impact on the probability of a female deer becoming pregnant. Table 3 shows the statistical significance of all the interaction terms tested. These regression coefficients were for mean-centered and scaled covariates.

The model was then optimized by removing insignificant interaction terms. Table 3 shows that maternal age and deer population density, as well as maternal age and County LCU, were statistically significant, and so the final, optimized model contained only those interaction terms along with all the individual covariates. The regression coefficients, 95% credibility intervals, and interpretations in the optimized model are provided in Table 4.

The intercept of the optimized regression model had a coefficient of 2.16 (95% CI: 0.66 to 3.63). This model had a Tjur’s R^2^ value of 0.63. Subtracting the null model’s R^2^ from the optimized model’s R^2^ yielded a value of 0.36, showing that the covariates in the optimized model explained an additional 27% of the variance in the data. However, 37% of the data variance was not accounted for by these models, indicating there are other covariates relevant to pregnancy probability.

## 4. Discussion

Examination of reproductive data in 9783 wild female Illinois white-tailed deer—the largest dataset of its kind reported so far—found no significant relationship between the probability of pregnancy and chronic wasting disease infection status (Table 4). While it may seem like a lack of effect of CWD on pregnancy would be good news, the finding that infected females are not impeded in their reproductive abilities by CWD infection means that breeding may be an important time for direct disease transmission between males and females. Infected females also risk transmitting the pathogen to their fawns through nursing, grooming, and other activities that involve close physical contact. This could be a source of infection early in life that would restrict the animal’s life expectancy and may explain the detection of CWD in some fawns, which are usually not tested for the disease due to its long incubation period. Transmission of CWD during the first few weeks of life would allow the infection time to reach a detectable level by the time that deer are being harvested (Fall) or culled (Winter). Even if transmission does not occur from female to fawn, previous studies conducted in mule deer [55] and white-tailed deer [33] have found that CWD-infected female deer may have lower fawn recruitment rates than their CWD-negative counterparts, which could contribute to disease-related population declines. Another consequence is that if CWD-infected females are just as able to get pregnant as CWD-negative females, it means they are competing with healthy females for mates and exposing males and their fawns to the pathogen. From a herd-health standpoint, it would be preferable for CWD infection to render females less reproductively fit, particularly if infected animals demonstrated more avoidant behaviors and less overall movement than uninfected animals. However, this is not the case, at least in white-tailed deer in northern Illinois, USA.

CWD’s lack of significance might be because many of the deer included in this study were within the incubation period of CWD, during which the physical consequences of CWD, such as wasting and neurodegenerative observable signs, have yet to develop. It is possible that symptomatic females would be affected, but additional data would need to be collected from deer in that stage of the disease to evaluate pregnancy. However, the findings of this study align with the literature from other species, which suggest that pregnancy is not severely limited by malnutrition or loss of body condition [33,36]. This conclusion is further supported by the omission of maternal body weight from the models and the observation that, despite increasing CWD prevalence in affected counties [43], deer populations continue to increase in the state of Illinois [10]. It should be noted that even though maternal weight lacks statistical significance, collecting maternal body weight should continue, when possible, as this variable remains of interest and could reveal different relationships as more data become available.

There was statistical significance in the relationships between pregnancy probability and other environmental and individual deer covariates. The day of the fiscal year on which the deer was culled had the simplest interpretation, since it was not part of an interaction term. The positive relationship between the day of the fiscal year and pregnancy probability means that pregnancy is easier to detect as the culling season progresses, with the fetuses in pregnant deer culled in March being larger and easier to identify than fetuses of deer culled in January. Interpretation of the relationship between maternal age and pregnancy probability was more complicated due to age being categorical and part of two significant interaction terms; however, general insights can still be gained. First, it can be observed that yearlings (1–2 years old) are 39.35% less likely than adults (2+ years old; the category of comparison) to be pregnant, and fawns (<1 year old) were 98.17% less likely (Table 4). This has been reflected in past literature about maternal age and reproduction, and shows that while it is possible for fawns to become pregnant, it is less common [9].

The maternal age category of “fawns” (8–10 months old) had a significant interaction with deer population density, so that female fawns living in areas with a high deer population density were 83.8% less likely to become pregnant than fawns living in areas with lower deer densities, possibly due to competition with their older counterparts for mates. County deer land cover utility (LCU) scores also had a significant impact on the relationship between fawns and pregnancy, with fawns living in areas with higher County LCUs—corresponding to more abundant and better habitat—305.52% more likely to become pregnant than female fawns in areas with lower County LCUs (Table 4). A better habitat is directly linked to better health and body condition, which are crucial for reproductive activity in fawns [14].

The relative magnitudes of the model terms involving fawns show that County LCU had the greatest impact on reproduction in this age group, followed by deer population density (Table 4). It may be interesting to note that County LCU is not statistically significant on its own, nor is it significant in its interaction with yearling as an age class (Table 4). It is thus the interplay between deer fawns and their habitat that matters for pregnancy. This may mean that deer habitat is abundant enough in northern Illinois such that there is little competition between female deer for space and resources, since if habitat quality and availability were limited in the area of study, yearlings and adults should also be affected by County LCU. Further research is needed to investigate these age-based relationships with environmental factors in order to gain a deeper understanding of how deer interact with their landscape.

A last observation of interest was that no interaction terms involving yearlings were significant, showing that yearlings and adults were equally sensitive to habitat (County LCU) and deer population density (Table 4). This could be useful in future modeling applications examining reproductive outcomes in wild white-tailed deer and may justify collecting separate data for fawns and yearlings, rather than grouping those age classes together under a label like “juvenile”.

The main limitation of this study is that the population examined was restricted to northern Illinois, USA. Expanding this analysis to other states with CWD would help identify which trends are shared across different geographies and which are specific to Illinois. Another limitation is that Tjur’s R^2^ of 0.63 indicates that there are covariates that influence pregnancy probability that were missing from this analysis. Several covariates could have influenced pregnancy probability if that data were available, including, but not limited to, whether the female deer had been pregnant in past fiscal years; contact rates with other female deer (as competition) and with males (as breeding partners); genetics; affliction with other infectious diseases, chronic conditions, or injuries; the health and fertility of the mothers of these female deer; and deer density in and near the female’s home range. Identification and inclusion of these, and other, covariates in future models would help us gain a deeper understanding of how CWD and other variables impact pregnancy in white-tailed deer. It should also be mentioned that the datasets used in this study did not include any information about the strain of CWD or the genetics of the female deer, both of which may influence disease vulnerability and progression. Overall, further evaluation and quantification of the effect of CWD on reproduction could impact adaptive management strategies for CWD in affected areas.

## Figures and Tables

**Figure 1 pathogens-14-00786-f001:**
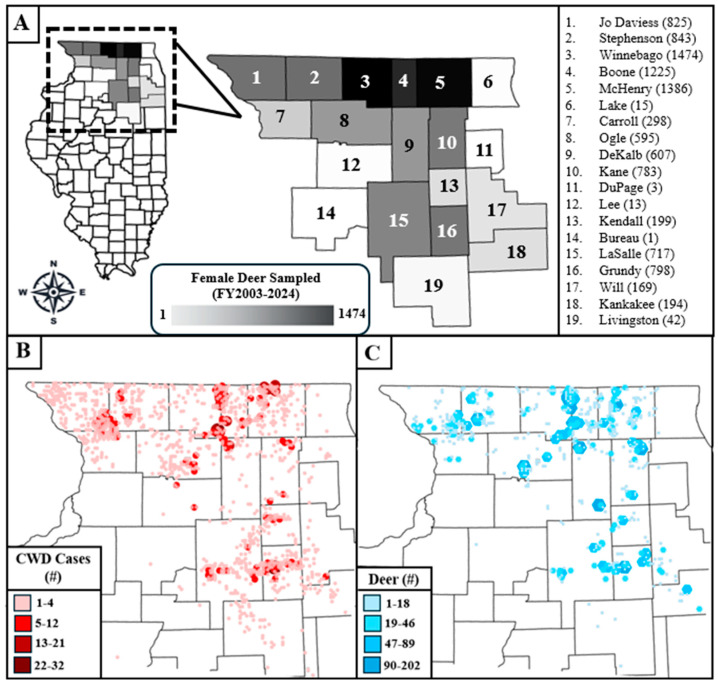
Maps of the study area in northern Illinois. (**A**) shows the counties included in the analysis, with the number of female deer that were culled and examined for pregnancy in parentheses beside the county name. (**B**) shows the cumulative number of chronic wasting disease (CWD) cases by TRS (township, range, and section) for all mortality sources between fiscal years 2003 and 2024 in the study area. (**C**) shows the total number of female deer included in this analysis by TRS. The fiscal year is the time between July 1st of one calendar year and June 30th of the next.

**Table 1 pathogens-14-00786-t001:** Covariates used in the Bayesian multilevel logistic regression quantifying their impact on the probability of pregnancy in northern Illinois white-tailed deer, USA, between fiscal years 2003–2024. A fiscal year is the time between July 1 of one calendar year and June 30th of the following calendar year.

Covariate	Definition	Units	Spatial Scale	Data Range	
				Weight	No Weight
Maternal age	Age of female deer when culled, grouped into fawns (<1 year old), yearlings (1–2 years), and adults (2+ years).	-	-	fawn, yearling, or adult ^2^	fawn, yearling, or adult
Maternal weight	Mass of female deer when culled.	kg	-	19.4 to 86.10	-
Maternal chronic wasting disease (CWD) status	CWD infection status of female deer.	-	-	0 (−) or 1 (+)	0 (−) or 1 (+)
Day of fiscal year	Day of fiscal year (starts July 1st).	day	-	201 to 272	196 to 275
Deer land cover utility index ^1^	Score given to a spatial area for its quality and quantity of deer habitat.	-	TRS	1141 to 10,806	231 to 10,973
			Township	305 to 10,072	305 to 10,072
			County	1014 to 8146	1014 to 8742
Deer population density	Deer removed by hunting in a county and fiscal year, divided by county area.	deer/km^2^	County	0.17 to 2.18	0.04 to 2.73

^1^ Deer land cover utility (LCU) data can be obtained from the Illinois Data Bank (https://doi.org/10.13012/B2IDB-0160590_V3). ^2^ Because animals are culled in the winter and spring, they are between 3 and 5 months older than animals harvested in the fall hunting season. Therefore, fawns are 8–10 months old, yearlings are 1–2 years old, and adults are older than 2 years.

**Table 2 pathogens-14-00786-t002:** Statistical significance of maternal age and weight in models using different combinations of county-level deer population and land cover utility (LCU) scores at the TRS, township, and county levels. A covariate was considered significant if its 95% credibility interval did not contain 0.

Covariate	Model				
	1 (LOOIC = 2346.7)	2 (LOOIC = 2498)	3 (LOOIC = 2332.6)	4 (LOOIC = 2368)	5 (LOOIC = 2229.6)
	Deer Population Density	TRS LCU	Township LCU	County LCU	Deer Population Density + TRS LCU
Age (fawn)	−1.41 (−2.3, −0.6)	−2.23 (−3.3, −1.1)	1.93 (−2.9, −0.9)	−1.37 (−2.2, −0.5)	−2.37 (−3.4, −1.3)
Age (yearling)	−0.25 (−0.6, 0.1)	−0.32 (−0.7, 0.1)	−0.26 (−0.7, 0.1)	−0.23 (−0.6, 0.1)	−0.33 (−0.7, 0.1)
Maternal weight	−0.15 (−0.7, 0.4)	−0.06 (−0.7, 0.6)	0.07 (0.6, 0.7)	−0.15 (−0.7, 0.4)	−0.04 (−0.7, 0.6)

**Table 3 pathogens-14-00786-t003:** Statistical significance of the interaction terms in the multilevel Bayesian logistic regression of pregnancy probability. An interaction was deemed significant if its 95% credibility interval did not contain 0. Covariates and interaction terms are classified as statistically significant (xxx) or not significant (-).

Interaction Term		95% Credibility Interval		
Covariate 1	Covariate 2	Lower Bound	Upper Bound	Significant
Maternal age (fawn)	CWD	−0.51	2.19	-
	Day of fiscal year	−0.34	0.29	-
	Deer population density	−2.47	−1.18	xxx
	County LCU ^1^	0.83	2.00	xxx
Maternal age (yearling)	CWD	−0.04	3.09	-
	Day of fiscal year	−0.62	0.13	-
	Deer population density	−0.11	1.33	-
	County LCU ^1^	−0.76	0.55	-
Deer population density	County LCU ^1^	−1.48	0.57	-

^1^ Deer land cover utility (LCU) scores.

**Table 4 pathogens-14-00786-t004:** Regression coefficients and interpretation of the pregnancy probability model. These coefficients were converted to aid in interpretation by applying an equation—(EXP (coefficient) − 1) × 100—to obtain the percent change in pregnancy probability when the covariate goes from a low to a high value. Change in pregnancy probability is only calculated for statistically significant covariates. Statistical significance was defined as the regression coefficient having a 95% credibility interval that did not contain 0.

Covariate	Regression Coefficient	95% Credibility Interval		Change in Pregnancy Probability (%)
		Lower Bound	Upper Bound	
Maternal age (fawn)	−4.00	−4.20	−3.81	−98.17%
Maternal age (yearling)	−0.50	−0.69	−0.30	−39.35%
CWD	−0.11	−0.54	0.34	-
Day of fiscal year	0.72	0.57	0.87	105.44%
Deer population density	1.67	0.41	2.89	431.22%
County LCU ^1^	−0.94	−2.15	0.30	-
Maternal age (fawn): Deer population density	−1.82	−2.46	−1.18	−83.80%
Maternal age (yearling): Deer population density	0.59	−0.13	1.32	-
Maternal age (fawn): County LCU ^1^	1.40	0.82	1.99	305.52%
Maternal age (yearling): County LCU ^1^	−0.10	−0.78	0.56	-

^1^ Deer land cover utility (LCU) scores.

## Data Availability

All data used in this study’s analyses are available at the Illinois Data Bank (DOI: https://doi.org/10.13012/B2IDB-6395643_V1).

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
