# Peer review of "Relationship Between Chronic Wasting Disease (CWD) Infection and Pregnancy Probability in Wild Female White-Tailed Deer (*Odocoileus virginianus*) in Northern Illinois, USA"

_pathogens, 2025, doi:10.3390/pathogens14080786_

Round 1
Reviewer 1 Report
Comments and Suggestions for Authors
This paper makes retrospective use of two long-running data sets to assess potential relationships between CWD and pregnancy in whitetails harvested for other purposes. The ms is extremely well written and authors have a good grasp and use of previous literature. It was a pleasure to read this ms. Conclusions are well within the scope of the data and the analyses. One could perhaps go one step further - and interpret the lack of association between CWD and pregnancy as implying that infected & uninfected does seem equally attractive and receptive to bucks, that CWD infection did not appear to translate into differences in behavioural response or scent attractors that was off-putting to bucks.
A few minor text edits:
-line 42: ... fronts, including their contribution ....
- l 50: change varies to differs
- l 57: successful
- l 97/98: possible due to sample bias associated with increased vulnerability ...
Author Response
-line 42: ... fronts, including their contribution ....
The word "from" was changed to "including" (line 44).
l 50: change varies to differs
"Varies" was changed to "differs" (line 50).
l 57: successful
"Success" - a mistype - was corrected to "successful" (line (53).
l 97/98: possible due to sample bias associated with increased vulnerability ...
Sentence was edited to say "possibly due to the sampling bias associated with increased vulnerability..." (lines 97-98).
Reviewer 2 Report
Comments and Suggestions for Authors
Mateus-Pinilla and colleagues present an article entitled “Relationship between chronic wasting disease (CWD) infection and pregnancy probability in wild female white-tailed deer (Odocoileus virginianus) in northern Illinois, USA”.
Chronic Wasting Disease (CWD) is part of the prion diseases, which are neurodegenerative diseases affecting mammals. CWD affects cervids. It first appeared in Colorado in the 1960s and has been spreading across North America, where it is now out of control. CWD is also present in South Korea and Scandinavia. The etiological agent is a host protein, the prion protein, whose conformation shifts into a pathological, aggregated, misfolded form. Different phenotypes have been described and are related to different prion strains, encoded in the structure of the assemblies.
The article focuses on the potential impact of CWD on the pregnancy probability in wild female white-tailed deer in northern Illinois. It uses records from the Illinois Department of Natural Resources, which has been monitoring CWD in wildlife since 2003.
The article is well-written and easy to follow, although some repetitions throughout the manuscript could be avoided, especially in the materials and methods section.
Here are my comments and concerns:
- At the end of the introduction, a brief mention of the conclusions is missing.
- Do the authors have access to the CWD strain affecting each animal to investigate potential differences between strains in terms of pregnancy impact?
- Do the authors have access to the genotype of each animal to investigate if some genotypes at positions 95 or 96, for example, could have some impact on the studied question?
- Since CWD has a long incubation period, the age of the animal is a crucial factor in the appearance of clinical signs. Did the authors have access to a more precise estimation of the age of the animals? Splitting the adult category into several age subcategories might be informative and highlight some differences.
- The reader would benefit from a figure showing the distribution of CWD cases and another one with the distribution of pregnant deer. Are the CWD cases homogeneously distributed in the different counties? In other words, is the percentage of CWD cases over the culled deer similar in the different counties? If not, could this lead to a bias in the study by taking into account areas with fewer CWD cases in the population, thereby diluting the effect that the authors tend to analyze?
- Data presented in Table 2 show that in the case of CWD as covariate 2, 0 is in the 95% credibility interval. However, 0 is really close to the lower bound, especially for yearlings. How clear-cut is it that CWD does not affect pregnancy? Deer population density and areas seem to affect pregnancy. In this context, could an inhomogeneous distribution of CWD in the region prevent a significant result?
- In the first paragraph of the results (page 6, lines 232-238), the authors do not show or present their results, just their analysis and conclusion. The authors have no choice but to trust the authors. Data should be included to support the authors’ claims.
- Could the authors propose other covariates that may explain their results to enrich their discussion (page 8, lines 274-276, and page 9, line 352)?
- On page 8, lines 286-288, the message is not clear. Do the authors mean that prions intend to do things for their proliferation? Could the authors clarify their message?
Minor remarks:
- The versions of R and the “brms” package used to make the statistical analyses are not mentioned.
- Tables would be easier to read with some horizontal lines delimiting each row.
Author Response
At the end of the introduction, a brief mention of the conclusions is missing.
A summary of the conclusions was added to the end of the Introduction (lines 135-136).
Do the authors have access to the CWD strain affecting each animal to investigate potential differences between strains in terms of pregnancy impact?
We do not have any data on the strain of CWD in these deer. This was mentioned as a limitation at the end of the Discussion (lines 427-429).
Do the authors have access to the genotype of each animal to investigate if some genotypes at positions 95 or 96, for example, could have some impact on the studied question?
We do not have any data on the genotypes of the deer in this study. This was mentioned as a limitation at the end of the Discussion (lines 427-429).
Since CWD has a long incubation period, the age of the animal is a crucial factor in the appearance of clinical signs. Did the authors have access to a more precise estimation of the age of the animals? Splitting the adult category into several age subcategories might be informative and highlight some differences.
The numerical age of the deer is sometimes estimated, but this data is not available for all deer, whereas the variable with the 3 age categories is. For this reason, the authors chose to use the age categories and not the numerical age to allow inclusion of many more data points. Future analyses could use numerical age instead. The reason for using a categorical age was mentioned in the text (lines 182-183).
The reader would benefit from a figure showing the distribution of CWD cases and another one with the distribution of pregnant deer. Are the CWD cases homogeneously distributed in the different counties? In other words, is the percentage of CWD cases over the culled deer similar in the different counties? If not, could this lead to a bias in the study by taking into account areas with fewer CWD cases in the population, thereby diluting the effect that the authors tend to analyze?
We added two maps to the original Figure 1 to show the cumulative number of CWD cases, as well as the number of female deer included from the study area. The map in Figure 1B shows the cumulative number of CWD cases and the map in Figure 1C shows the total number of female deer examined for pregnancy that were included in the study. The original Figure 1 is included as Figure 1A, which some reformatting. Comparison between Figures 1B and 1C show that the sample size of female deer is, generally, proportional to the number of CWD cases, so that the areas in which CWD is more prevalent had more female deer included in the analysis. We believe this observation - that the number of deer included is proportional to CWD cases - allays the concern about a dilution effect. In addition, the inclusion of the spatial levels "County/Township/TRS" in the model should account for differences in sampling between different geographic locations. Discussion of these maps was added in a paragraph immediately preceding Figure 1.
Data presented in Table 2 show that in the case of CWD as covariate 2, 0 is in the 95% credibility interval. However, 0 is really close to the lower bound, especially for yearlings. How clear-cut is it that CWD does not affect pregnancy? Deer population density and areas seem to affect pregnancy. In this context, could an inhomogeneous distribution of CWD in the region prevent a significant result?
There are a few factors at play here. One factor is that the regression coefficients are from covariates that have been standardized and standardized covariates tend to yield smaller regression coefficients, so this is not unusual. Another factor is the use of Bayesian methods, which more fully characterize and account for model/parameter uncertainty, thereby increasing confidence in the regression coefficients. Also, as discussed in the response to the prior comment, deer sampling was proportional to CWD cases, so that while CWD cases are clustered on the landscape, more deer are sampled in places with more CWD. Lastly, the dataset used in this analysis contained 5,673 pregnant deer and 4,110 deer that were not pregnant, which provides a balanced dataset that's less likely to be influenced by chance or biases.
In the first paragraph of the results (page 6, lines 232-238), the authors do not show or present their results, just their analysis and conclusion. The authors have no choice but to trust the authors. Data should be included to support the authors’ claims.
A new table was added (now Table 2) that provided the results of the models examining maternal age and weight in the same models. The findings were discussed in the paragraph before the new table. Subsequent table numbers were also updated.
Could the authors propose other covariates that may explain their results to enrich their discussion (page 8, lines 274-276, and page 9, line 352)?
Potential additional covariates that may contribute to explaining pregnancy rates were added to the end of the Discussion.
On page 8, lines 286-288, the message is not clear. Do the authors mean that prions intend to do things for their proliferation? Could the authors clarify their message?
It was clarified that the differences may be the behavior of infected animals, rather than implying that the prion itself is driving these differences (lines 288-290).
The versions of R and the “brms” package used to make the statistical analyses are not mentioned.
The versions of R (line 125) and the "brms" package (line 196) were added to the text.
Tables would be easier to read with some horizontal lines delimiting each row.
Horizontal lines were added to the tables.
Reviewer 3 Report
Comments and Suggestions for Authors
In this manuscript, Mori et. al. used a multilevel Bayesian logistical regression model to quantify pregnancy probability with a subset of covariates; CWD infection of the dam was the priority. The results presented show that CWD infection does not affect the ability for a dam to become pregnant while other covariates do. The manuscript is well written. There are 4 comments for the authors to consider.
Comment #1:
The authors compiled and presented 21 years of data in a model system yet provided no reference for the need of this analysis other than it will help managers and this has not yet been done. Is there data from the IDNR that demonstrates that recruitment rates are down? Or that population rates are down? The data on CWD prevalence, expansion, and herd density in these areas must be known. Would the authors consider correlating hard data with your modeling system? This would greatly strengthen the manuscript.
Comment #2:
The authors found no correlation with maternal CWD infection and the ability to become pregnant, then state in the abstract (line28-29) and discussion (280-282) major concerns about the role reproduction has on CWD transmission. Can the authors please expand upon why they present this viewpoint, and more specifically, what aspects of reproduction are of concern?
Comment #3:
The authors present in their findings that 37% of the data variance is not accounted for and that ‘other’ covariates are affecting pregnancy probability. Can the authors expand upon what other covariates they would consider for future analysis?
Comment #4:
The findings of this manuscript, along with other published work, appears to demonstrate that becoming pregnant is not a problem for CWD infected dams. There is still concern about recruitment rates and viable fetuses as a cause for population decline, which have also been published. Would the authors consider expanding upon this talking point in the discussion?
Minor points:
In the methods section, the number of pregnant and non-pregnant female deer numbers assessed in this study are provided, then deer population density is provided as a covariate. Does this include both male and female deer, and if so, what is the ratio of males to females that make up the density?
Line 141: The acronym TRS is used but not defined, then defined on line 152.
Author Response
Comment #1:
The authors compiled and presented 21 years of data in a model system yet provided no reference for the need of this analysis other than it will help managers and this has not yet been done. Is there data from the IDNR that demonstrates that recruitment rates are down? Or that population rates are down? The data on CWD prevalence, expansion, and herd density in these areas must be known. Would the authors consider correlating hard data with your modeling system? This would greatly strengthen the manuscript.
This is a good question, thank you. We clarified that the reason for this study is that there is concern that CWD is reducing reproductive rates in Illinois white-tailed deer, as has been observed or predicted in other states, and we conducted our analysis because no one has yet evaluated this question in Illinois. Our interest is in whether CWD has impacted pregnancy to date. We added this information to the end of the Introduction.
Comment #2:
The authors found no correlation with maternal CWD infection and the ability to become pregnant, then state in the abstract (line28-29) and discussion (280-282) major concerns about the role reproduction has on CWD transmission. Can the authors please expand upon why they present this viewpoint, and more specifically, what aspects of reproduction are of concern?
In the Abstract, it was clarified that the concern related to a lack of effect of CWD on pregnancy is because the equal ability of infected or non-infected females to breed may make reproduction an important part of CWD spread via direct transmission from infected females to their fawns. This point was reiterated and elaborated upon in the beginning of the Discussion.
Comment #3:
The authors present in their findings that 37% of the data variance is not accounted for and that ‘other’ covariates are affecting pregnancy probability. Can the authors expand upon what other covariates they would consider for future analysis?
Potential additional covariates that may contribute to explaining pregnancy rates were added to the end of the Discussion.
Comment #4:
The findings of this manuscript, along with other published work, appears to demonstrate that becoming pregnant is not a problem for CWD infected dams. There is still concern about recruitment rates and viable fetuses as a cause for population decline, which have also been published. Would the authors consider expanding upon this talking point in the discussion?
We added discussion of these aspects to the first paragraph of the Discussion.
Minor points:
In the methods section, the number of pregnant and non-pregnant female deer numbers assessed in this study are provided, then deer population density is provided as a covariate. Does this include both male and female deer, and if so, what is the ratio of males to females that make up the density?
It was clarified that the deer population density includes both sexes and all age categories, and that the sex ratio is roughly 50/50.
Line 141: The acronym TRS is used but not defined, then defined on line 152.
The definition of TRS was moved to the first occurrence of the acronym.
st occurrence of the acronym.
Reviewer 4 Report
Comments and Suggestions for Authors
A study of incubation times is more difficult to conduct on a herd of wild deer, but a comparison can be made with scrapie.
In sheep flocks with scrapie, farmers observed:
- increased ewe prolificacy
- a gradual decrease in the incubation period of the disease (the prion gradually adapting to the herd's genetics)
- maternal transmission of scrapie was observed with much shorter incubation times than usual (personal observation: 6 months of incubation)
Author Response
A study of incubation times is more difficult to conduct on a herd of wild deer, but a comparison can be made with scrapie.
In sheep flocks with scrapie, farmers observed:
- increased ewe prolificacy
- a gradual decrease in the incubation period of the disease (the prion gradually adapting to the herd's genetics)
- maternal transmission of scrapie was observed with much shorter incubation times than usual (personal observation: 6 months of incubation)
A section was added to the Introduction that mentions parallels in prion-disease impacts on cattle and sheep.
Round 2
Reviewer 2 Report
Comments and Suggestions for Authors
The authors answered all my questions and concerns.
Author Response
Thank you for taking the time to review our paper.
Reviewer 3 Report
Comments and Suggestions for Authors
Previous Reviewer Comment #1:
Is there data from the IDNR that demonstrates that recruitment rates are down? Or that population rates are down?
This is a good question, thank you. We clarified that the reason for this study is that there is concern that CWD is reducing reproductive rates in Illinois white-tailed deer, as has been observed or predicted in other states, and we conducted our analysis because no one has yet evaluated this question in Illinois. Our interest is in whether CWD has impacted pregnancy to date. We added this information to the end of the Introduction.
Reviewer Response: Thank you to the authors for their response. While the additions bring clarity to the original question, a quick search on-line showed the white-tailed deer populations in Illinois have been steadily increasing over the past few years. Taken at face value, one would assume that pregnancy is not being affected by CWD. This study does demonstrate that this is the case. This reviewer’s initial question pertained to why no known, informative data was used to confirm/support the results of this study? CWD prevalence rates for both state and counties of Illinois I’m sure is available. Deer populations for the state and specific regions I’m sure are available. For future studies, this type of information would greatly support the questions being asked and provide more relevance for expanding the findings to other regions or states.
Reviewer Comment: Line 91; a space is needed between references and the word ‘and’
Reviewer Comment: All other issues have been adequately addressed and the manuscript should be accepted for publication.
Author Response
We added a sentence in the Discussion mentioning that despite an increase in CWD prevalence in affected counties over time, the Illinois deer population continues to increase as well (lines 373-375). We also fixed the error on line 91.